# Secondary Metabolites Isolated from *Artemisia afra* and *Artemisia annua* and Their Anti-Malarial, Anti-Inflammatory and Immunomodulating Properties—Pharmacokinetics and Pharmacodynamics: A Review

**DOI:** 10.3390/metabo13050613

**Published:** 2023-04-29

**Authors:** Lahngong Methodius Shinyuy, Gisèle E. Loe, Olivia Jansen, Lúcia Mamede, Allison Ledoux, Sandra Fankem Noukimi, Suh Nchang Abenwie, Stephen Mbigha Ghogomu, Jacob Souopgui, Annie Robert, Kristiaan Demeyer, Michel Frederich

**Affiliations:** 1Laboratory of Pharmacognosy, Department of Pharmacy, Center of Interdisciplinary Research on Medicine (CIRM), University of Liege, 4000 Liège, Belgium; 2Laboratory of In Vitro Toxicology and Dermato-Cosmetology (IVTD), Department of Analytical, Applied Chemometrics and Molecular Modeling (FABI), Faculty of Medicine and Pharmacy, Vrije Universiteit of Brussel, 1050 Ixelles, Belgium; 3Laboratory of Pharmacochemical and Natural Pharmaceutical Substances, Doctoral Training Unit in Health Sciences, Faculty of Medicine and Pharmaceutical Sciences, University of Douala, Douala P.O. Box 2701, Cameroon; 4Molecular and Cell Biology Laboratory (MCBL), Department of Biochemistry and Molecular Biology, Faculty of Science, University of Buea, Buea P.O. Box 63, Cameroon; 5Embryology and Biotechnology Laboratory, Université Libre de Bruxelles, 1050 Brussels, Belgium; 6Epidemiology and Biostatistics Unit (EPiD), Institute of Clinical and Experimental Research (IREC), UCLouvain, 1200 Brussel, Belgium

**Keywords:** malaria, artemisinin, metabolites, *Artemisia afra*, *Artemisia annua*

## Abstract

There are over 500 species of the genus *Artemisia* in the Asteraceae family distributed over the globe, with varying potentials to treat different ailments. Following the isolation of artemisinin (a potent anti-malarial compound with a sesquiterpene backbone) from *Artemisia annua*, the phytochemical composition of this species has been of interest over recent decades. Additionally, the number of phytochemical investigations of other species, including those of *Artemisia afra* in a search for new molecules with pharmacological potentials, has increased in recent years. This has led to the isolation of several compounds from both species, including a majority of monoterpenes, sesquiterpenes, and polyphenols with varying pharmacological activities. This review aims to discuss the most important compounds present in both plant species with anti-malarial properties, anti-inflammatory potentials, and immunomodulating properties, with an emphasis on their pharmacokinetics and pharmacodynamics properties. Additionally, the toxicity of both plants and their anti-malaria properties, including those of other species in the genus *Artemisia*, is discussed. As such, data were collected via a thorough literature search in web databases, such as ResearchGate, ScienceDirect, Google scholar, PubMed, Phytochemical and Ethnobotanical databases, up to 2022. A distinction was made between compounds involved in a direct anti-plasmodial activity and those expressing anti-inflammatory and immunomodulating activities or anti-fever properties. For pharmacokinetics activities, a distinction was made between compounds influencing bioavailability (CYP effect or P-Glycoprotein effect) and those affecting the stability of pharmacodynamic active components.

## 1. Introduction

If one were to nominate the most infectious parasitic diseases in the world, malaria would be at the top of the list, as it is a disease of public health importance with serious socio-economic and human consequences. Malaria is a devastating disease, affecting over 241 million people worldwide annually, with 627,000 deaths [1] There were about 14 million more cases of malaria and 69,000 more deaths in 2020 than there were in 2019, most of which were in Sub-Saharan African countries. Approximately 95% of the world’s malaria burden is concentrated in 11 countries, with 10 of them being in Sub-Saharan Africa with 96% of the death rate; over 80% of these are among children under the age of five and pregnant women [1] This is due to increasing anti-malarial drug resistance to the chemotherapy employed at present. The situation is further aggravated by not only Histidine-Rich Protein-2 gene (HRP2 gene) deletion, making diagnosis problematic, but also by mosquito resistance to insecticides and invasive vector species (*Anopheles stephensi*) in the Horn of Africa [1]. The fight against this life-threatening disease is one of the goals of the World Health Organization’s (WHO) Global Technical Strategy to reduce its transmission and help create a world that is free of malaria by 2030. The terrestrial ecosystem has been a source of many lead compounds that have undergone chemical derivatization and drug development to produce anti-microbial molecules, including those used to fight malaria [2]. Numerous studies have been conducted on natural products such as crude extracts or pure compounds isolated from medicinal plants. Ethnobotanical and ethnomedical alternatives for the management of health problems, including malaria, have been practiced for many years and are used by 80% of the global population in primary health care [3,4]. These alternative remedies have provided leads for the development of drugs that are useful in therapeutics such as those in used in Western medicine. A well-known example of the seminal contribution of ethnomedicine to the treatment of malaria in the modern medicinal way includes artemisinin, which is isolated from *A. annua* [5]. Following the isolation of this well-known anti-malarial molecule, intensive research on the phytochemistry of this species has resulted in the isolation of almost 600 secondary metabolites in recent decades [6]. The emergence of resistance to all current malaria drugs including artemisinin has increased the used of *A. annua* and *A. afra* by the population, especially in areas of high endemicity. *A. annua* has been used in China since ancient times as a herbal product taken in the form of tea infusions or decoction for the alleviation of intermittent fevers associated with malaria, amongst other diseases [7]. *A. annua* is documented in the *Pharmacopeia of the People’s Republic of China* and is now domesticated in other parts of Asia, Europe, Australia, and America, where it is used to fight other ailments [7,8,9,10]. However, the WHO has cautioned against the use of non-pharmaceutical sources of artemisinin because of the risk of delivering sub-therapeutic doses of artemisinin that could potentiate anti-malarial drug resistance [11]. Hence, the emergence of *A. afra* which contains negligible or no artemisinin content but kills malaria parasites [12] including the gametocytes [13,14]. *A. afra* has a long history of use as a herbal product in South Africa for the treatment of ailments, including malaria [15] and it is cultivated across the African continent for the management of malaria and other diseases [9,12]. About 400 secondary metabolites have been isolated from *A. afra* with a wide range of biological activities against various diseases, including malaria [9]. Being a member of the Asteraceae family, the genus *Artemisia* is one of the largest and most dispersed genera, with over 500 species, which are predominantly located in different geographical regions due to their ability to thrive, survive and persist in almost all habitat types [11]. The ethnobotanical and ethnomedicinal importance, potential phytotherapeutic application and traditional use of this plant genus have been reviewed by Parada et al. [16] and Wright et al. [17].

In this review, the main objective is to discuss the main secondary metabolites isolated from *A. afra* and *A. annua* and their anti-malarial and anti-inflammatory potentials, with an emphasis on their pharmacokinetics and pharmacodynamic importance.

## 2. Methodology

### Literature Search

A thorough review of the literature in web databases relating to secondary metabolites isolated from *A. annua* and *A. afra* and their anti-malaria, anti-inflammatory and immunomodulating properties, with a focus on their pharmacokinetics and pharmacodynamics properties, was performed. A literature search of the research in web databases was conducted, and data up to 2022 were collected from published journal articles in international scientific databases, including Ethnomedicinal, ResearchGate, ScienceDirect, Google scholar, PubMed, Phytochemical and Ethnobotanical, etc., reporting on the traditional use, phytochemistry, anti-malaria, anti-inflammatory, immunomodulating, pharmacokinetics and pharmacodynamics of *A. annua* and *A. afra* and their secondary metabolites. The following key terms were employed for the literature search: Malaria/burden/*A. annua*/*A. afra*, botanical description, geographical distribution/Secondary metabolites/Asteraceae family/Anti-malarial, anti-inflammatory/Immunomodulation/Indigenous use/Herbal Remedies/Ethnobotany/Ethnopharmacology/Ethnomedicine/Ethnomedicinal/Ethnopharmaceutical/phytochemistry/Cytochrome/Cytochrome P450/Pharmacology/pharmacokinetics, pharmacodynamics, bioavailability/Resistance/Artemisinin Combination Therapy. About 200 published research articles were studied, and chemical structures of bioactive compounds were drawn using a scientifically accepted program ChemDraw.

## 3. Results

### 3.1. Botanical Description and Distribution

*Artemisia annua* L. is also referred to as sweet wormwood, sweet annie, and annual wormwood (English) or qinghao (Chinese). According to the *Pharmacopeia of the People’s Republic of China* [18], a plant containing artemisinin in the Asteraceae family is termed quinghao, meaning “the dry” above-ground parts. *A. annua* is a large, annually grown, highly aromatic herbaceous herb that can grow up to 2 m tall, with a single stem covered with fine grey-green hairs. It has aromatic leaves which are deeply dissected.

Despite it being widely distributed throughout the world due to its ability to treat a wide range of diseases, traditionally, *A. annua* originated in China [19]. Just like most of the members of the Asteraceae family, *A. annua* is highly adaptable, growing in almost all hostile environments, such as forest margins, roadsides, hillsides, dry river valleys, forest meadows, rocky slopes, semiarid climates, and grasslands. However, it was reported that the best growing condition for this plant is in a humid and subtropical monsoon climate with average temperatures of 17.6–28.4 °C [20]. *A. annua* grows in most parts of the temperate and subtropical parts of the central, southern, and eastern parts of Europe and Asia. Extending to the south-eastern part of Asia, this plant is also seen growing across the Mediterranean and northern parts of Africa. It has also been spotted in the United States, North America, and Canada [19], Figure 1.

Additionally, referred to as African wormwood, *A. afra* (Jac. ex Willd) is described as having different common names in different societies [22]. Being an indigenous herb to Africa, *A. afra* grows across the African continent in areas extending from Cederberg Mountains in the Cape to tropical East Africa and continuing through to the north, including Ethiopia [22]. Described as a condiment [18], it has been spotted in almost every comer in the world due to its high medicinal potential (Figure 1). *A. afra* grows up to a height of at least 2 m. It is a woody perennial herb with finely divided oval-shaped leaves, and it has an aromatic smell, and alternately arranged, silver-grey leaves in the adaxial region and light green leaves in the axial region [15].

### 3.2. Phytochemistry of A. afra and A. annua

For decades, the phytochemical investigation of *A. annua* has been at the forefront of research due to the isolation of artemisinin, a potent anti-malarial compound belonging to the sesquiterpene lactone class [5]. Further research has led to the isolation of almost six hundred secondary metabolites, the majority of which are terpenoids, polyphenols and coumarins, as summarized in the Appendix A. There have been several publications on the phytochemistry of *A. annua* over the previous decade. Table 1 summarizes the secondary metabolites isolated from both plant species over the previous decade. As such, several new secondary metabolites have been identified and characterized, with most of them belonging to the monoterpenes and sesquiterpenes classes and a few polyphenols.

For instance, Qin et al. [26]. isolated five new sesquiterpene compounds from *A. annua*: Arteannoid A, which is a sesquiterpenoid dimer composed of two cadinene sesquiterpenoid units; Arteannoids B and C, which are rearranged heterodimers of the cadinene sesquiterpene and one phenylpropanoid unit and two new rearranged cadinene sesquiterpenoids (Arteannoides D and E). In addition, thirteen new sesquiterpenoid compounds (Arteannoides F–R) were isolated from the aerial part of *A. annua* [39]. Arteannoid H is a new eudesmane-type sesquiterpenoid [26]. A comparative phytochemical investigation of two chemotypes of *A. annua* (high- and low-artemisinin-producing chemotypes, HAP and LAP, respectively) via UPLC, GC-MS and NMR also lead to the isolation and characterization of twenty-six novel compounds belonging to monoterpenes and sesquiterpenes classes. Nineteen of these secondary metabolites were highly oxygenated amorphane sesquiterpenes, which are the most diverse and abundant subclass [23]. De Magalhães et al. [34]. also confirmed the presence of phenolic acid (Rosmarinic acid) in the leaves of *A. annua*. Mouton and Van der Kooy [32] isolated two compounds (melilotosides) with anti-typhoid potentials. Goel et al. [40] also characterized nine new aliphatic compounds from the aerial part of *A. annua*.

As it is a member of the same genus, *A. afra* shares almost the same pattern of secondary metabolites, which is dominated by terpenoids, polyphenols and coumarins, as summarized in the Appendix A. While *A. annua* cultivars have been cultivated and selected for their high artemisinin content, *A. afra* is devoid of this potent sesquiterpene lactone [41]. However, other studies have shown that there are cultivars of *A. afra* with negligible contents of artemisinin [13]. Research on the phytochemistry of *A. afra* has led to the isolation of almost five hundred secondary metabolites comprising monoterpenes, diterpenes, triterpenes, sesquiterpenes, guaianolides, glaucolides, coumarins and polyphenols (Appendix A). Surprisingly, a very limited amount of work on phytochemistry has been conducted on *A. afra* over the previous decade despite its multi-biological activity against infections [15]. However, Emmanuel et al. [36] isolated five new sesquiterpenes from this species and two other compounds (*p-hydroxy acetophenone* and 2,4-dihydroxy-6 methoxy acetophenone). Sotenjwa et al. [24] also isolated two new compounds, a flavonoid (rutin) and a coumarin (scopolin), from this species.

### 3.3. Anti-Malarial Properties, Anti-Inflammatory and Immunomodulating Effects of A. annua and A. afra

According to the Chinese culture, *Artemisia annua* has been known to alleviate of fevers since the second century [7,9]. It is a highly efficacious medicinal plant against malaria due to the presence of artemisinin and is traditionally used in infusions or decoctions for the treatment of malaria in China [37]. Ethnopharmacological studies for the use of *A. annua* in Traditional Chinese Medicine (TCM) prompted the isolation and characterization of artemisinin [5], which is a sesquiterpene with remarkable efficacy against the erythrocytic stages of the malaria parasite. Several in vitro and in vivo studies on the crude plant material and/or tea infusions have been reported, showing good anti-plasmodial activities [42,43]. For example, Diawara et al. [43] confirmed the in vitro anti-malarial activity of aqueous and hydroalcoholic crude extracts of *A. annua*, with an IC_50_ of 4.95 nM for chloroquine resistance against *P. falciparum*. Recent investigations on the effect of tea infusions of *A. annua* on pre-erythrocytic and erythrocytic stages showed growth inhibition of various *Plasmodium* species in a dose-dependent manner [9]. In addition, Snider et al. [41] confirmed the anti-parasitic activity of a tea infusion with *A. annua* against the early and late stages of *P. falciparum* gametocytes, as well as against the asexual forms. Table 2 summarizes the reports on anti-malarial properties.

Several clinical trials of *A. annua* either in tea infusions or as a whole-plant powder taken in the form of a capsule have also been reported, showing the effectiveness of this plant as a traditional remedy against malaria [12,43]. For example, a clinical study was conducted in the Democratic Republic of Congo using an infusion of high artemisinin (0.5–0.75%) cultivar (*A. annua* cv *Artemi*) at two doses (5 g of herb in 1 L of water and 9 g/L) in separate groups, and it indicated 77% and 70% cure rates, respectively, after seven days of treatment [64]. However, there were high levels of recrudescence [64]. Blanke et al. [65], also reported a 70% cure rate with *A. annua* tea following a 7-day treatment, with a high level of recrudescence after day 28 in a small, randomized, double-blinded trial. The high level of recrudescence could be due to the rapid clearance of artemisinin, as it is metabolized by cytochrome P450 (CYPs) enzymes.

Cytochrome P450 (CYPs) is a superfamily of hemeprotein enzymes present in the livers of humans, whose function is to metabolize xenobiotics. These enzymes are involved in the first-pass effect of many drugs, including artemisinin, which is actively metabolized by CYP2B6 and CYP3A4 enzymes [52]. Medicinal plants and or pure molecules with safety profiles that can inhibit cytochrome P450 enzymes, especially CYP2B6 and CYP3A4, which are responsible for the metabolism of artemisinin, can be utilized as a traditional remedy or as modern therapy in combination with artemisinin, respectively, to fight malaria. In an in vitro investigation of the effect of an infusion of *A. afra* and *A. annua* on human hepatic CYP2B6 and CYP3A4 enzymes, both plant extracts inhibited the activity of both enzymes [52]. It was also shown that *A. annua* is a potent inhibitor of CYP3A4 [23]. This could be due to the presence of molecules other than artemisinin within the plant matrix.

Despite it being devoid of artemisinin or having a negligible artemisinin content [13,66]. *A. afra* has equally demonstrated good anti-plasmodial properties both when it is used in infusions or in crude extracts in vitro, in vivo and in clinical studies (Table 2). *A. afra* is native to South Africa and has a long history of use as a traditional remedy against coughs, colds, fever, and malaria [15]. Tea infusions of *A. afra* are used to treat malaria traditionally [66]. Crude extracts of *A. afra* (ethanol, hexane, and dichloromethane extracts) collected from 5 different countries in Africa demonstrated anti-plasmodial activity against chloroquine-resistant *P. falciparum* strains [47]. Report on the antimalarial activities of *A. afra* both in vitro, and in vivo as well as clinical studies has been reviewed by du Toi and van der Kooy [67]. However, there are recent investigations on the anti-plasmodial activities of tea infusions of *A. afra* in vitro. For example, Tea infusions of *A. afra* cultivars with traces of artemisinin (19 nM artemisinin) and without detected levels of artemisinin demonstrated activity against the early-stage gametocytes associated with altered morphology of the gametes [41]. This level of activity was shown to be higher in cultivars containing traces of artemisinin [41]. In the same study, activity against the asexual forms of *P. falciparum* was reported [41]. Furthermore, the *A. afra* tea infusion demonstrated anti-plasmodial activity against the pre-erythrocytic and erythrocytic stages of *P. falciparum* in vitro in a dose-dependent manner [9].

Attention has been paid to *A. afra* and *A. annua* either in infusions or crude extracts for the investigation of anti-plasmodial properties over the years. However, there are several other *Artemisia* spp. for which a potential anti-malarial activity has been reported, either in vitro or in vivo. These include *Artemisia abyssinica*, *A. apiacea*, *A. gorgonum*, *A. absinthium*, *A. vulgaris*, *A. lancea*, *A. nilagirica*, *A. abrotanum*, *A. japonica*, *A. nilegarica*, *A. ciniformis*, *A. bienis*, *A. turanica*, *A. indica* and *A. maciverae*. An overview is given in Table 2.

During a malaria attack, xanthin oxidase (XO) is upregulated, leading to the production of reactive oxygen species (ROS), which trigger inflammation [68,69,70]. Ty et al. [71] showed a correlation between elevated plasma-level activities of XO and cytokine production in malaria patients. Pro-inflammatory cytokines together with other cell mediators are responsible for the pathogenesis of inflammatory diseases. Medicinal plants and pure molecules that can inhibit the production of these pro-inflammatory cytokines can be utilized as a traditional remedy or as a modern therapy for the supportive treatment of these diseases. In addition to their anti-malarial properties, *A. annua* and *A. afra* have also been reported to exhibit anti-inflammatory properties. For instance, hot water extracts of *A. annua* prepared in a method similar to traditional ways demonstrated anti-inflammatory properties by modulating pro- and anti-inflammatory cytokines [34,72]. At the molecular level, Abate et al. [73] confirm the anti-inflammatory potential of leaf extracts of *A. annua* with hydro-ethanol for the inhibition of TNF-α gene expression. Additionally, the anti-inflammatory potential of *A. afra* has been shown to significantly inhibit the production of nitric oxide (NO) and interleukin 6 (IL-6) in a dose-dependent fashion in several investigations [74].

There are other species of medicinal plants (Figure 2) that belong to the genus Artemisia with reported anti-inflammatory potentials, including *Artemisia vulgaris*, *A. absinthium*, *A. apiacea*, *A. dracunculus*, *A. campestris*, *A. vestila*, *A. inculta*, *A. montana*, *A. scoporia*, *A. vulgaris*, *A. sieversiana*, *A. frigida*, *A. copa*, *A. argyi* and *A. princeps*.

### 3.4. Anti-Malarial Properties, Anti-Inflammatory and Immunomodulating Effects of Secondary Metabolites

Following the isolation of almost 600 and 400 secondary metabolites from *A. annua* [6] and *A. afra* [66] respectively, both plant species have served as a reservoir for bioactive molecules against different ailments, including malaria. Some of the isolated metabolites have been shown to have a direct anti-malarial property, while others act by increasing the bioavailability of active components against malaria. Artemisinin is one of the potent molecules with direct anti-malarial activity. However, following the emergence of resistance by the parasite against artemisinin, recent investigations on the biological activity of other secondary metabolites against malaria in the quest for the discovery of new molecules have gained attention in recent decades. The presence of these metabolites could potentialize artemisinin or reduce the risk of developing resistance.

In this context, the presence of flavonoids may be a relevant key factor, broadening the pharmacological properties of Artemisia-based preparations. For instance, in an in vitro investigation of quercetin against malaria parasites, the authors showed that quercetin has moderate, direct anti-plasmodial activity, with an IC_50_ value of 19.31 ± 1.26 μM, and a high selectivity index [76]. In the same study, anti-malarial properties of quercetin in *P. berghei*-infected mice demonstrated that it performs chemo-suppressive activity in a dose-dependent manner. Jansen et al. [77], further confirmed the moderate anti-plasmodial properties of quercetin in vitro, with an IC_50_ value of 9.5 μg/mL (31.4 μM). Additionally, three flavonoids; rutin, rhamnetin and quercetin, alongside derivatives of quercetin, inhibited the growth of laboratory strains and field isolates of *P. falciparum* in an in vitro investigation [78]. Using an experimental set-up to determine the anti-malarial activity of flavonoids from *A. annua*, Willcox, et al. [79] also showed that chrysosplenetin, chrysosplenol-D, cirsilineol and artemetin have direct anti-plasmodial activities in vitro (with an IC_50_ range of 26–65 μmol/L) and that these molecules could also potentiate the effect of artemisinin on *P. falciparum*. Artennuic acid and artennuin B were also shown to have weak direct anti-malarial activity [80]. In an in vivo study conducted to assess the anti-malarial properties of apigenin, a flavonoid common to both *A. afra* and *A. annua*, this compound was shown to significantly suppress the growth of *P. berghei* in infected mice in a dose-dependent way [81]. In addition, Fallatah and Georges [82] reported that active efflux of glutathione from mature erythrocytes was induced by apigenin through an erythrocytes ATP-binding cassette subfamily C member 1 (ABCC1)-mediated mechanism. This effect may lead to oxidative stress in *P. falciparum*-infected erythrocytes, and therefore, the inhibition of parasite proliferation, as confirmed in an in vivo study by Amiri et al. [81]. Additionally, kaempferol (a flavonoid) was shown to inhibit the growth of *P. berghei* in a dose-dependent manner [83]. These researchers could show that kaempferol has a curative, prophylactic and chemo-suppressive effect on malaria in vivo. However, Jansen et al. [77], reported a lack of anti-plasmodial properties of kaempferol in vitro, although the researchers isolated the molecule from medicinal plants other than those of the Asteraceae family. Kaempferol also acts as an inhibitor of glycogen synthase kinase-3β (GSK3β) of the malaria parasite [84]; hence, it blocks erythrocyte invasion by the parasite. Barlianna et al. [85] confirmed the moderate anti-plasmodial properties of a derivative of kaempferol (kaempferol-3-*O*-rhamnoside) in *P. falciparum*, with a reported IC_50_ value of 106 μM. Figure 3 summarizes structures of compounds with direct or indirect anti-malaria properties.

While some components show a direct anti-malarial activity, others may act indirectly by synergistically increasing the bioavailability of active compounds. One mechanism by which this may be achieved is through the inhibition of cytochrome P450 enzymes, which are involved in the rapid metabolism of active compounds and have an immunomodulating effect. In an in vivo study carried out by Li et al. [86] to investigate the effect of the interaction between artemisinin, artennuin B, artennuic acid and scopoletin on *P. yoelii*-infected mice, the researchers showed an increase in the level of exposure of artemisinin, which enhanced the mean plasma concentration and led to improved anti-malarial activity. In their findings, the potency of their combination (four-compound therapy) was almost four times higher compared to that of pure artemisinin. Artennuin B, artennuic acid and scopoletin were believed to have improved the bioavailability of artemisinin [87], which is highly metabolized by CYP450 enzymes, including CYP3A4 [87]. Additionally, artennuin B has been demonstrated to be a potent inhibitor of CYP3A4 [88], and this effect could enhance the mean plasma concentration of artemisinin, leading to improved bioavailability, and therefore, an increase in activity. Thujone, norilodol and artemisia ketone, on the other hand, have been reported to be potent inhibitors of CYP2B6 [88,89].

Several of these compounds with direct or indirect anti-malarial properties have also shown anti-inflammatory properties. In addition to its anti-malarial properties described above, kaempferol also has anti-inflammatory properties [90]. In an investigation carried out by Kim et al. [91] on the anti-inflammatory properties of artemisinin extracted from *A. annua* using different solvents, artemisinin was also found to have an inhibitory effect on lipopolysaccharide-induced nitric oxide (NO), prostaglandins and proinflammatory cytokine production, such as IL-1β, IL-6 and IL-10. Additionally, artemisinin was found to downregulate pro-inflammatory cytokines, and hence, ameliorate inflammation [92] in an in vivo investigation. Furthermore, artemisinin was found to be an inhibitor of TNF-α expression [89]; hence, this demonstrates its anti-inflammatory potential at the molecular scale. Additionally, artennuin B isolated from *A. annua* has a strong inhibitory effect on the production of cytokines, such as IL-1β, IL-6 and TNF-α [93]. This could further explain the anti-inflammatory potential of *A. annua*. Table 3 summarizes compounds isolated from *A. annua* and *A. afra* with direct or indirect anti-plasmodial and anti-inflammatory properties.

Quercetin showed a GSK3β-mediated cytokine-modulating effect by regulating pro- and anti-inflammatory cytokine levels in *P. berghei*-infected mice [76]. In this investigation, quercetin was also found to significantly reduce the levels of TNF-α and IFN-Y and raise the levels of IL-10 and IL-4 [76]. In a study carried out by Min et al. [98] to determine the anti-inflammatory effect of eupatilin and jasceosidin on carrageenan-induced inflammation in mice, both molecules were found to inhibit the expression and activation of cyclooxygenase (COX)-2 and nuclear factor kappa β, respectively. The same authors further reported a reduction in TNF-α and IL-1 β, as well as prostaglandin E2 (PGE2), levels in mice by eupatilin and jasceosidin. The researchers isolated these molecules from *Artemisia princeps*; however, these molecules have also been isolated from *A. annua* and *A. afra*, and they are also present in other *Artemisia* species. Qin et al. [26], also reported the anti-inflammatory properties of two newly rearranged heterodimers of cadinene sesquiterpene and one phenylpropanoid unit (Arteanoids B and C).

### 3.5. Pharmacokinetics and Bioavailability of Secondary Metabolites of A. annua and A. afra

While pharmacokinetics describes the absorption, distribution, metabolism, and elimination of a drug or pharmakon, bioavailability, on the other hand, describes the fraction of the absorbed drug that reaches the systemic circulation, and it varies between compounds, both of which influence the observed therapeutic or biological responses [101]. The pharmacokinetics and bioavailability of artemisinin and derivatives have been more extensively studied either in vitro, in vivo or in clinical studies than other metabolites in this plant species have. This is probably due to its excellent anti-malarial properties, and it is also a drug that is used in therapy. Several pharmacokinetic studies have been conducted on artemisinin administered as a single compound or in a plant matrix, such as in whole-plant powder or as an infusion, both of which may influence the pharmacokinetics of artemisinin. For example, in a pharmacokinetic study of artemisinin following the oral administration of a traditional preparation of *A. annua*, it was found that artemisinin is absorbed faster from herbal tea preparations than it is from capsules (from 30 min to 2.3 h) [102]. However, both formulations had a similar bioavailability. Desrosiers et al. [103] reported increases in the serum levels of artemisinin when it is administered as a whole plant as compared to that when the capsule of artemisinin is administered, indicating that the plant matrix has a positive impact on the bioavailability of artemisinin. In a multi-component assessment of the pharmacokinetics parameters of *A. annua* in rats, Fu et al. [104] showed that scopoline, scopoletin, rutin, chrysosplenol D, casticin and three sesquiterpenes (arteannuin B, dihydroartemisinic acid and artemisinic acid) were detected in rat’s plasma post-oral administration. In the same study, the authors concluded that chrysosplenol D and casticin were rapidly absorbed with shorter half-lives (t1/2, 2.68 ± 3.62 h and 0.33 ± 0.07 h, respectively) as compared to the speed of the absorption of scopoletin, which has a longer half-life (t1/2, 6.53 ± 1.84 h). The long half-life of scopoletin in vivo could explain its effect on the modulation of cytochrome P450 enzymes and the modulation of pro- and anti-inflammatory cytokines. In the same study, scopoletin, casticin, artemisinin and chrysosplenol D were metabolized by phase II enzymes. Following the intravenous administration of eupatilin in mice, it was found that eupatilin was poorly adsorbed due to its low systemic exposure [105]. Eupatilin was rapidly metabolized by phase II enzyme into eupatilin-7-glucuronide (E-7-G), and both molecules were distributed in the intestine, liver, and kidneys. It was confirmed by the authors that eupatilin has a shorter half-life (t1/2, 0.29 h) as compared to that of its metabolite, E-7-G, which has a longer half-life (t1/2, 4.15 h) and might be responsible for the potency of eupatilin in vivo. Poor absorption and a high biliary elimination rate limit the bioavailability of quercetin. However, the absorption of quercetin depends on the chemical structure of the molecule. Quercetin aglycon is mainly absorbed in the small intestine and stomach [106], while quercetin rutinoside is absorbed in the colon [92]. Erlund et al. [107] also confirmed the absorption of quercetin aglycon in the gastrointestinal tract in clinical studies. The results from an in vitro incubation of human small microsomes with quercetin aglycon show that it is highly metabolized by phase II enzymes in the small intestine [108]. Apigenin, a natural flavonoid common in the Asteraceae family and found in other plant species, has a poor systemic availability due to its low solubility in water and lipids. After a single oral administration in rats, apigenin was found widely distributed throughout the body (liver, kidneys, RBCs, urine, and intestine) within 10 days post-administration, and it was excreted mainly in urine [109]. Apigenin is extensively metabolized by phase I and phase II enzymes, making its bioavailability very low [109,110]. The results of the oral and intravenous administration (400 mg/kg and 50 mg/kg, respectively) of casticin in rats showed rapid distribution and elimination with a longer half-life (t1/2, 36.48 min ± 7.24) for the oral administration as compared to those of intravenous administration (t1/2, 20.86 min ± 2.02) [111]. The absolute bioavailability (45.5 ± 11.0%) was high, which is an important factor for its clinical application [111]. Barve et al. [112] reported a low bioavailability (approximately 2%) of kaempferol due to its extensive first pass-effect metabolism in rats after oral administration. In the same study, the plasma concentration of kaempferol indicated a rapid absorption (tmax,1–2 h) with a half-life of 3–4 h and with a large distribution (8–12 L/h/kg) and a high rate of clearance (approximately 3 L/h/kg). Zabela et al. [113] also reported the rapid clearance (4.40–6.44 L/h/kg) and extremely short half-life (t1/2, 2.93–3.79 min) of kaempferol after intravenous administration in rats. In a study carried out to determine the effect of plant matrix on the uptake of luteolin from aqueous solutions of *A. afra*, luteolin aglycone and luteolin-7-0-glucoside were more rapidly absorbed than when they were administered as pure solutions in Caco-2 cells after 1 h exposure. These results suggest that plant matrix may have a positive effect on the bioavailability of the flavonoid and hence greater in vivo potency [114].

### 3.6. Toxicity of Crude Extracts of A. annua and A. afra and Their Secondary Metabolites

Pre-clinical toxicity is often required for new substances and new medicinal products, which contain herbs with no traditional history of use. According to the WHO, traditional use refers to documentary evidence that a substance has been used for a specific health-related or medicinal purpose over three or more generations. The guidelines of the WHO state that if a herbal product has a long history of traditional use without signs of toxicity (harmful effects), it should not be restricted, and they maintains it stand that no pre-clinical toxicity investigation is required [115]. *Artemisia annua* and *Artemisia afra* have a long history of use as a traditional remedy in Southeast Asia and South Africa, respectively, for the alleviation of fevers related to malaria, amongst other diseases, for generations. However, several toxicity studies (cytotoxicity, acute and chronic toxicity) have been conducted to determine the safety of *A. annua* and *A. afra*, including a couple of their secondary metabolites. For example, Motshudi et al. [116] showed that crude extracts of *A. afra* prepared using chloroform, ethanol and water were relatively non-toxic to vero cells, with LC_50_ values > 400 g/mL, as compared to that of the positive control (doxorubicin), which was cytotoxic, with an LC_50_ value of 57.83 ± 3.02 mg/mL. Loggenberg et al. [117] also confirmed the non-cytotoxic effect of aqueous extract of *A. afra*, with IC_50_ > 350 μg/mL. In the same study, *A. annua* was confirmed to be non-cytotoxic, with IC_50_ > 550 μg/mL, to vero cells. In addition to in vitro toxicity assays used to assess the safety profiles of *A. annua* and *A. afra*, several in vivo studies (acute and chronic studies) have been conducted to confirm the claims observed in, in vitro studies. For instance, in an investigation carried out by Mekenon et al. [118] to evaluate the toxicity of aqueous extracts of *A. afra* leaves on brain, heart and suprarenal glands in Swiss albino mice via an acute and sub-acute test, the researchers showed that leaves extract of *A. afra* is relatively safe for mice, with an LD_50_ > 5000 mg/kg body weight. In the sub-acute study, the microscopic examination of brain, heart and suprarenal glands showed no signs of toxicity in all the treatment groups. Kane et al. [119] also confirmed the safety of *A. afra* leave extracts in vivo by determining its pharmaco-toxicological effects after acute oral administration in mice at doses of 1000, 2000 and 2500 mg/kg as per body weight. An LD_50_ > 2500 mg/kg body weight, no signs of toxicity and no effect on the levels of alanine transaminase (ALT) and aspartate transaminase (AST) were reported and compared to those of the control in the study. Additionally, following the oral administration of hydro-ethanolic plant extract of *A. annua* by Swiss mice, the results showed no lethality or toxic reactions, with LD_50_ > 5000 mg/kg body weight [120]. The absence of toxicity symptoms suggests that *A. annua* is non-toxic and is well tolerated [120]. According to the classification of Loomis and Haye, substances with LD_50_ values between 500 and 5000 and between 5000 and 15,000 mg/kg body weight are regarded as being slightly toxic and practically non-toxic, respectively. The safety profiles of *A. afra* was further confirmed by Mukinda et al. [121] by using an aqueous extract of *A. afra* in acute and chronic tests on mice and rats, respectively. The pharmaco-toxicological effects of the extract were determined after intraperitoneal and oral administration. The researchers reported LD_50_ values of 2450 and 8960 mg/kg body weight for intraperitoneal and oral administration, respectively. Both methods of administration were safe as the LD_50_ values fell within the cut-off mark, following the guidelines of Loomis and Haye [122]. After a chronic investigation of rats administered oral doses of *A. afra* extract (0.1 or 1 g/kg/day), the rats survived 3 months of dosing, and LD_50_ > 1000 mg/kg body weight was reported. There were no significant changes in their general behavior and hematological and biochemical parameters experienced, except for a transient decrease in AST activity level. No significant changes were observed in organ weights, and the histopathological results showed a normal profile, suggesting no morphological alterations. The above findings clearly support the use of *A. annua* and *A. afra* as a traditional remedy in the fight against infections.

Several compounds isolated from *A. annua* and *A. afra* have demonstrated good biological activities against different ailments, including malaria, with several toxicity studies having been conducted to determine their safety profiles. For example, in an investigation to determine the cytotoxic effect of dietary flavones on normal trout liver cell lines, the authors of [123] showed that while methylated flavones (5,7-Dimethoxyflavone and 3′,4′-dime-thoxyflavone at concentration 25 μM) are non-cytotoxic even at higher concentrations after 24 h of exposure, the hydroxylated flavones (Chrysin, quercetin, apigenin and luteolin), especially chrysin, were cytotoxic, with a reported IC_50_ value of 2 μM. However, in a sub-chronic toxicity study carried out by Yao et al. [124], following the daily oral administration of chrysin (1000 mg/kg), the compound significantly decrease body weight, liver weight was increased, there were significant alteration in the hematology, and an LD_50_ value of 4350 mg/kg was reported. The results from a sub-chronic toxicity study of quercetin examined in male and female CD2F1 mice administered at 62, 125 and 250 mg/kg indicated that quercetin is safe to CD2F1 mice, as they showed no signs of toxicity, and no changes in hematological and histopathological parameters were observed [125]. Lucida et al. [126] also confirmed the safety of quercetin in an acute test, with an LD50 value > 1600 mg/kg. Several studies have been conducted to investigate the toxicity of volatile components of *A. annua* and *A. afra*. For instance, in an investigation to evaluate the toxic effects of α-terpinene (a monoterpene common to both species) in the liver tissue of rats intraperitoneally administered at doses of 0.5, 0.75 and 1.0 mL/kg for 10 days, Baldissera et al. [127] showed that α-terpinene induces oxidative stress and has cytotoxic and genotoxic effects on the liver tissue, involving caspase activation. This was due to increased levels of reactive oxygen species, alanine aminotransferase, aspartate aminotransferase increased and glutathione S-transferase. There are several reviews that have detailed the toxicity of volatile components of both plants [89,128,129].

### 3.7. Resistance to ACTs and Broad-Spectrum Pharmacology of Artemisia-Based Samples

With the persistence of malaria as a devastating disease with high burden and mortality rates, the WHO recommends “Artemisinin Combination Therapy (ACTs)” as the first line of treatment (artemether + lumefantrine, artesunate + amodiaquine, artesunate + mefloquine, dihydroartemisinin + piperaquine or artesunate + sulfadoxine–pyrimethamine) [1]. However, there have been reported cases of resistance to ACTs, leading to treatment failures. Artemisinin resistance was documented in 2008 on the Thailand–Cambodia border after artesunate monotherapy [130]. However, retrospective analysis indicated that artemisinin resistance likely emerged in 2001, before the widespread deployment of ACTs in Cambodia [131]. The mechanism of resistance to artemisinin drugs is associated with point mutations in the Kelch 13 (K13) propeller gene. In Cameroon and other African countries (Central African Republic, Chad, Kenya, Madagascar, Malawi, Mali, Rwanda, etc.), non-synonymous K13 mutations have been reported, and the most frequent allele observed in Africa is A578S [132]. Further studies revealed that four mutations (Y493H, R539T, I543T and C580Y) of the K13-propeller gene in *P. falciparum* are associated with artemisinin resistance [132].

The emergence of resistance to ACTs has prompted the search for new strategies to fight against the disease. The optimal therapy for the management of the disease would consist of one that will target both gametocidal and clinical forms of the parasite. This will enable them to break the transmission cycle, and hence, it’s possible eradication. The isolation of gametocidal compounds from *A. afra* [14] and in vitro and in vivo investigations of the anti-plasmodial activity against the erythrocytic forms of malaria parasite [41,64], [133,134] as well as clinical results from trial studies, indicate that *A. afra* may be an important contributor in the fight against malaria. The presence of the plant matrix, as reported, may play an important part, as it improves the pharmacokinetics of molecules with direct anti-plasmodial properties and those with indirect anti-plasmodial properties [114]. This could be explained by the fact that other molecules present in the plant may limit the rate of the metabolism of compounds with direct anti-plasmodial properties by inhibiting the action of cytochrome P450 enzymes responsible for the metabolism of these compounds. This further demonstrates the potential of *A. afra* in the fight against malaria. Additionally, new drug combinations with molecules that could enhance the bioavailability of artemisinin, together with compounds that have direct anti-malarial properties, including those showing gametocidal effects, could function best in eliminating the disease. As an example, a drug combination or a standardized plant extract containing artemisinin (a potent anti-malarial molecule), artennuin B (a CYP3A4 inhibitor, but also showing a direct anti-malarial activity), quercetin (with direct anti-malarial activity and an inhibitor of thioredoxin reductase and anti-inflammatory), yomogiartemin (a gametocidal molecule) and thujone (has an direct anti-malarial effect and a CYP2A6 and CYP2B6 inhibitor) could be of potential interest in treating malaria.

## 4. Conclusions

Based on in vitro and/or in vivo studies and limited clinical trials, there are more than twenty different species of *Artemisia* with the anti-malarial/anti-inflammatory properties investigated. Research over the years in the search for new molecules showing anti-malarial activities has been focused on erythrocytic forms of the malaria parasite, with limited work having been conducted on the pre-erythrocytic and gametocidal forms.

In addition to artemisinin, several secondary metabolites present in different *Artemisia* spp. show direct anti-malarial activity. Other metabolites act indirectly by increasing the bioavailability of active components or by affecting other metabolic pathways that are important in sustaining the patient when they are fighting a malaria infection. A few of the isolated secondary metabolites with anti-plasmodial properties and immunomodulating effects, as well as anti-inflammatory properties, have been studied for their pharmacokinetics and pharmacodynamics properties in humans and their toxicological profiles. It would, therefore, be interesting to broaden the knowledge of Artemisia-based preparations by investigating their total pharmacological effectiveness in clinical trials in relation to their chemical composition. The focus should, therefore, be put on investigating the presence of molecules in addition to artemisinin, resulting in a broad pharmacological activity and effectiveness during different stages of the malaria parasite, especially the gametocidal forms. This would enable the reduction of the link between human and insect vectors, and hence, reduce malaria transmission.

## Figures and Tables

**Figure 1 metabolites-13-00613-f001:**
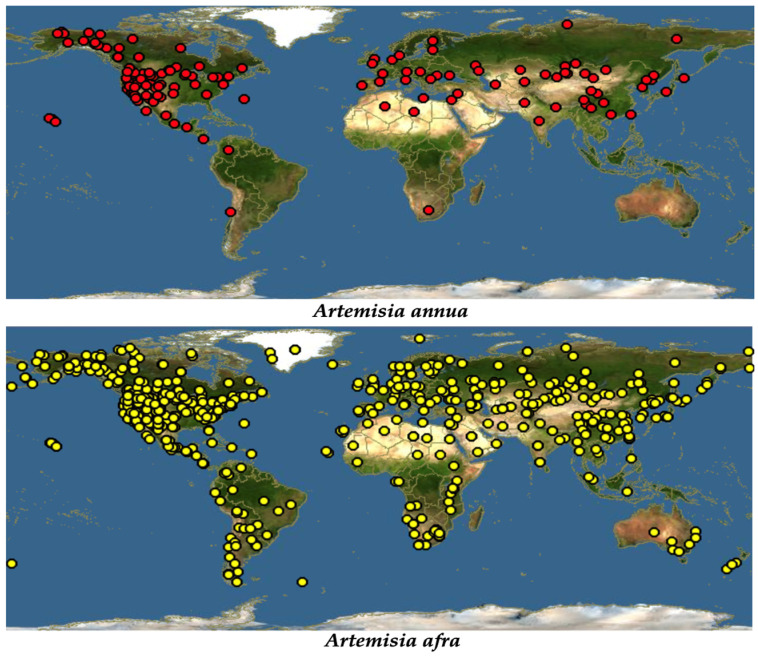
Global distribution of *A. annua* and *A. afra* [21].

**Figure 2 metabolites-13-00613-f002:**
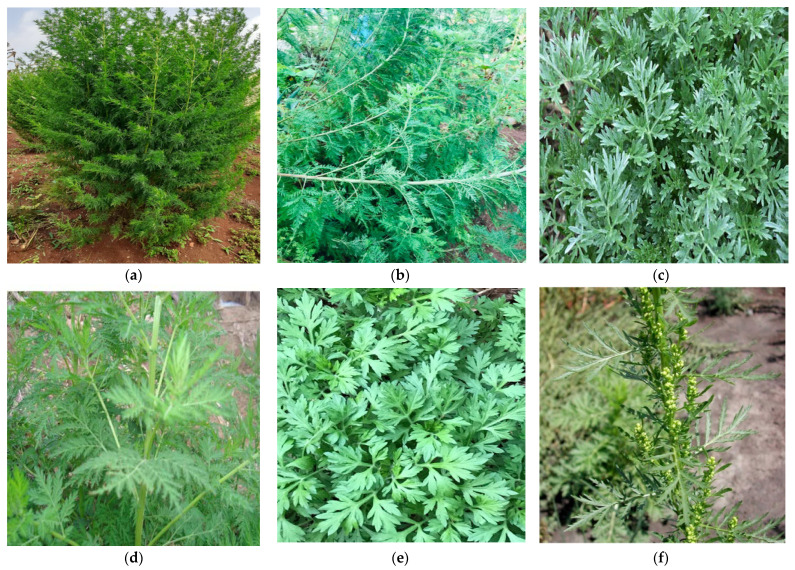
Field pictures of *Artemisia* spp. [75] with reported anti-plasmodial properties: (**a**) *A. annua*, (**b**) *A. afra*, (**c**) *A. absinthium*, (**d**) *A. vulgaris*, (**e**) *A. indica*, (**f**) *A. lancea*, (**g**) *A. japonica*, (**h**) *A. abrotanum nilagirica*, (**i**) *A. bienis*, (**j**) *A. nilagirica* and (**k**) *A. gorgonum*.

**Figure 3 metabolites-13-00613-f003:**
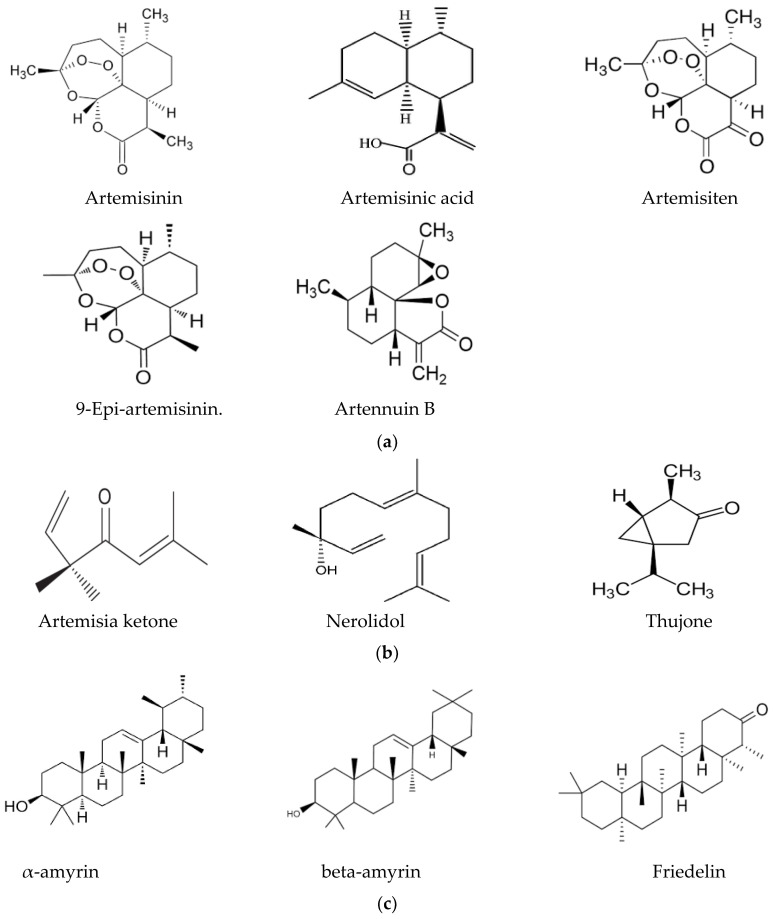
(**a**) Sesquiterpenes with direct anti-plasmodial activity. (**b**) Monoterpenes with direct anti-plasmodial activity. (**c**) Triterpenes with direct anti-plasmodial activity. (**d**) Polyphenols with direct anti-plasmodial activity.

**Table 1 metabolites-13-00613-t001:** Secondary metabolites isolated from *A. annua* and *A. afra* in the previous decade.

Plant	Compound Name	Classification	References
*Artemisia* *annua*	Abeo-amorphane sesquiterpene	Sesquiterpene	[23]
	(Z)-7-Acetoxy-methyl-11-methyl-3-methylenedodeca-1,6,10-triene	Sesquiterpenes	[24]
	Apigenin-6-C-hexoside-8-C-pentoside	Flavonoid	[10]
	Arteannoides A to E	Sesquiterpene	[25]
	Arteannoides F to R	Sesquiterpene	[26]
	Arteannoides U to Z	Sesquiterpenes	[25]
	Artemisinic acid, 6α-peroxy ester	Sesquiterpene	[23]
	Artemisiannuside A	Coumarin glycoside	[27]
	Artemanin A	Sesquiterpene	[28]
	Artemanin B	Sesquiterpene	
	Arteannuin P	Monoterpene	[23,29]
	Arteannuin Q	Monoterpene	
	Arteannuin S	Sesquiterpene	
	Arteannuin T	Sesquiterpene	
	Arteannuin U	Sesquiterpene	
	Arteannuin V	Sesquiterpene	
	Arteannuin W	Sesquiterpene	
	Arteannuin Y	Sesquiterpene	
	Arteannuin Z	Monoterpene	
	Cadinanolide	Sesquiterpene	[27]
	Caffeoylcoumaroyltartaric acid	Phenolic acid	[10]
	n-Cos-(Z)-9-enoic acid.	unsaturated fatty acid	[30]
	n-Cos-(Z)-10-enoic acid	unsaturated fatty acid	
	3-p-o-Coumaroyl-5-O-caffeoylquinic acid	Flavonoid	[10]
	Chrysoeriol rutinoside	Flavonoid	[31]
	*Cis*-melilotoside	Coumarin	[32]
	Dehydroarteannuin L	Sesquiterpene	[23]
	Deoxyartemistene	Sesquiterpene	
	Dihydroxy-dimethoxyl-O-hexoside	Sugar	[10]
	3,4-Dihydroxybenzyl 2′,3′,4′-trihydroxybenzoate	dihydroxybenzyl ester	[30]
	3,4-Dihydroxybenzyl 2′,3′,4′- trihydroxybenzoate 4,4′-β-D-dixylopyranoside	dihydroxybenzyl ester	
	4α, 5α-Epoxy-6α- hydroxyartemisinic acid methyl ester	Sesquiterpene	[23]
	3-(2-(2,5-dihydrofuran-3-yl) ethyl)-2,2-Dimethyl-4-methylenecyclohexan-1-one	Sesquiterpene	
	(R)-15,16-Didehydrocoriolic acid	-	[27]
	Eriodictyol-7-O-hexoside	Flavonoid	[10]
	*Epi*-11-hydroxy-arteannuin I	Sesquiterpene	[23]
	6,7-Epoxy-6,7-dihydro-β-farnesene	Monoterpene	
	n-Heptadecanyl-ß-D-glucopyronoside	aliphatic	[30]
		alcoholic glucoside	
	n-Heptadecanyl linoleate	unsaturated fatty acid	
	Homoeriodictyol	-	[27]
	5β-Hydroperoxy-eudesma-4(15),11-diene	Sesquiterpene	[23]
	7α-Hydroxy-artemisinic acid	Sesquiterpene	
	11-Hydroxy-arteannuin I	Sesquiterpene	
	6α-Hydroxy-arteannuin J	Sesquiterpene	
	(*E*)-7-Hydroxy-2,7-dimethylocta-2,5-dien-4-one	Monoterpene	
	(*E*)-7-Hydroperoxy-2,7-dimethylocta-2,5-dien-4-one	Monoterpene	
	6- Hydroxy-γ-humulene	Monoterpene	
	Isoarteannuin A	Sesquiterpene	
	Isodocosanol	Aliphatic alcohol	[30]
	Isorhamnetin-O-hexoside	Flavonoid	[10]
	Isononadecano	Aliphatic alcohol	[30]
	Luteolin 7-O-pentoside	Flavonoid	[10]
	1-Octacosanol	Aliphatic alcohol	[30]
	n-Octadecanyl n-octadec-9,12,-dienoate	unsaturated fatty acid	
	n-Nonacosanyl n-octadec-9,12-dienoate	unsaturated fatty acid	
	(±)-Qinghaocoumarin A	Coumarin	[33]
	Qinghaocoumarin B	Coumarin	
	Qinghaolignan A	Lignan	
	Qinghaolignan B	Lignan	
	Qinghaosu I and III	Sesquiterpene	
	Quinic acid	Organic acid	[10]
	Rosmarinic acid	Phenolic acid	[34]
	*Trans*-melilotoside	Coumarin	[32]
	5,7,4′-Trimethoxy-8,3″-dihydroxyflavone	Flavonoid	[23]
	Trimethoxy-coumarin	Coumarin	[35]
*Artemisia* *afra*	Artemin	Sesquiterpene	[36]
	Artesin	Sesquiterpene	
	Arabinose	Sugar	[37]
	Fucose	Sugar	
	1α,4α-Dihydroxybishopsolicepolide	Guaianolide	[14]
	2,4-Dihydroxy-6 methoxyacetophenone	-	[36]
	Galacturonic acid	Organic acid	[37]
	Galactose	Sugar	
	Glucose	Sugar	
	Glucuronic acid	Organic acid	
	*p*-Hydroxyacetophenone	-	[36]
	Mannose	Sugar	[37]
	Maritimin	Sesquiterpene	[36]
	4-O-Methylglucuronic acid	Organic acid	[37]
	Reynosin	Sesquiterpene	[36]
	Rhamnose	Sugar	[37]
	Rutin	Flavonoid	[36]
	Santolinifolide	Sesquiterpene	
	Santolinifolide A	Sesquiterpene	
	Scopolin	Coumarin	[24]
	Yomogiartemin	Guaianolide	[38]

**Table 2 metabolites-13-00613-t002:** *Artemisia* spp. with reported anti-malarial properties either in vitro, in vivo or in clinical trials.

Plant Species	Plant Part/Extract	Model	Stage of Parasite	IC_50_ or %Inhibition	References
*Artemisia absinthium*	Leaves/CF	In vitro	Erythrocytic	0.42 μg/mL	[44]
	Leaves/MeOH	In vitro	Erythrocytic	20.00 μg/mL	[45]
	Leaves/EtOH	In vivo	Erythrocytic	60% inhibition at 100 mg/Kg/day	[45]
*Artemisia abyssinica*	Leaves/PE	In vitro	Erythrocytic	2.10 μg/mL	[46]
*Artemisia afra*	Leaves/ethanolic	In vitro	Erythrocytic	2.66 μg/mL	[47]
	Leaves/hexanolic	In vitro	Erythrocytic	0.71 μg/mL	[46]
	Leaves/CF	In vitro	Erythrocytic	8.55 μg/mL	[48]
	Leaves/DCM	In vitro	Erythrocytic	6.58 μg Artemisinin/L	[43]
	Leaves/tea infusion	In vitro	Erythrocytic	100% inhibition of parasite growth at 4 g/L	[41]
	Leaves/tea infusion	In vitro	Pre-erythrocytic	100% inhibition of parasite growth at 4 g/L	[41]
	Leaves and twigs/tea infusion	In vitro	Erythrocytic	Significant decrease in parasitemia at 5 g/L	[41]
	Leaves and twigs/tea infusion	In vitro	Gametocytes	Significant decrease in gametocytaemia at 5 g/L	[41]
	Leaves/hydro-ethanol	In vitro	erythrocytic	0.46 μg/mL	[45]
	Leaves/DCM	In vivo	Erythrocytic	94.28% inhibition at a dose of 200 mg/kg body weight	[45]
*Artemisia armeniaca*	Aerial part/PE	In vitro	Erythrocytic	0.90 mg/mL	[49]
	Aerial part/DCM	In vitro	Erythrocytic	1.04 mg/mL	[49]
*Artemisia annua*	Leaves/twigs	Leaves/crude extracts	Erythrocytic	0.51, 0.52, and 1.11 μg/mL for W, B, and R respectively	[50]
	Leaves/tea infusion	Leaves/tea infusion	Pre-erythrocytic	85% inhibition of parasite growth at 10 g/L	[51]
	Leaves/tea infusion	In vitro	Erythrocytic	100% inhibition of parasite growth at 10 g/L	[51]
	Leaves and twigs/tea infusion	In vitro	Erythrocytic	Significant decrease in parasitemia at 5 g/L	[41]
	Leaves and twigs/tea infusion	In vitro	Gametocytes	Significant decrease in gametocytaemia at 5 g/L	[41]
	Leaves/tea infusion	In vitro	Erythrocytic	7.21 μg artemisinin/L	[43]
	Leaves/DCM	In vitro	Erythrocytic	3.79 μg artemisinin/L	[43]
	Leaves/methanol	In vitro	Erythrocytic	3.00 artemisinin/L	[43]
	Leaves/aqueous	In vitro	Erythrocytic	4.95 nM	[50,52]
	Leaves/ethanolic	In vivo	Erythrocytic	80% inhibition at 20 mg/kg artemisinin per day for 5 days.	[50,52]
	Leaves/ethanolic	In vitro	Erythrocytic	3.27 nM	[50,52]
	Leaves/hexane	In vivo	Erythrocytic	26% suppression of parasitemia after 4 days of treatment at 75 mg/kg/day of extract	[50]
	Leaves/tea infusion	In vitro	Erythrocytic	0.095 μg/mL	[53]
	Leaves/tea infusion	In vitro	Erythrocytic	1.11 μg/mL and 0.88 μg/mL for CQ-sensitive and CQ-resistant strains, respectively	[54]
	Leaves/DCM	In vivo	Erythrocytic	83.28% inhibition at a dose of 200 mg/kg body weight	[42]
	Leaves/tea	Clinical trial	Erythrocytic	92% inhibition of parasitemia at 5 g/L	[55]
	Leaves/Acetone	In vitro	Gametocidal	<10 μg/mL	[14]
	Leaves/DCM	In vitro	Erythrocytic	3.04 μg/mL	[46]
*Artemisia aucheri*	Leaves/DCM	In vitro	Erythrocytic	1.00 mg/mL	[51]
	Aerial part/DCM	In vitro	Erythrocytic	1.95 mg/mL	[49]
*Artemisia biennis*	Aerial part/DCM	In vitro	Erythrocytic	5.2 μg/mL	[56]
	Leaves/DCM	In vitro	Erythrocytic	0.78 mg/mL	[51]
*Artemisia gorgonum*	Aerial part/EtOH	In vitro	Erythrocytic	2.64 mg/mL	[57]
*Artemisia indica*	Leaves/hexane	In vitro	Erythrocytic	4.40 μg/mL	[46]
	Aerial part/Combination of EtOH, MeOH, PE (1:1:1)	In vivo	Erythrocytic	-	[58]
	Leaves/CF	In vitro	Erythrocytic	7.09 μg/mL	[59]
	Leaves/PE	In vitro	Erythrocytic	10.24 μg/mL	[59]
	Leaves/hexane	In vitro	Erythrocytic	9.88 μg/mL	[59]
	Leaves/MeOH	In vitro	Erythrocytic	5.76 μg/mL	[59]
	Leaves/EtOH	In vitro	Erythrocytic	11.37 μg/mL	[59]
*Artemisia judaica*	Leaves/EtOAC	In vitro	Erythrocytic	1.35 mg/mL	[60]
	Leaves/DCM	In vitro	Erythrocytic	9.02 mg/mL	[60]
*Artemisia roxburghiana*	Leaves/DCM	In vitro	Erythrocytic	1.93 mg/mL	[60]
*A. scoparia*	Leaves/DCM	In vitro	Erythrocytic	0.78 mg/mL	[51]
*Artemisia siebera*	Leaves/PE	In vitro	Erythrocytic	2.88 mg/mL	[60]
*Artemisia spicigera*	Leaves/DCM	In vitro	Erythrocytic	1.00 mg/mL	[51]
*Artemisia turanica*	Leaves/DCM	In vitro	Erythrocytic	0.92 mg/mL	[60]
*Artemisia turcomanica*	Aerial part/methanol	In vivo	Erythrocytic	82.40% inhibition at 500 mg/kg	[61]
	Leaves/DCM	In vitro	Erythrocytic	4.90 μg/mL	[46]
*Artemisia vulgaris*	Leaves/H_2_O	In vitro	Erythrocytic	20 μg/mL	[46]
	Leaves/Acetone	In vitro	Erythrocytic	1.9 μg/mL	[46]
	Aerial part/ethanol	In vivo	Erythrocytic	87.30% inhibition at 1000 mg/kg	[62]
	Leaves/combination of MeOH, PE and water (1:1:1)	In vivo	Erythrocytic	65.16% inhibition of parasitemia at 500 mg/kg	[63]

MeOH = methanol; EtOH = ethanol; CF = chloroform; PE = petroleum ether; DCM = dichloromethane; EtOAC = ethyl acetate; W = crude extract prepared from *A. annua* grown under white LED light (445 and 554 nm); B = crude extract prepared from *A. annua* grown under blue LED light (445 nm); R = crude extract prepared from *A. annua* grown under red LED light (660 nm); CQ = chloroquine.

**Table 3 metabolites-13-00613-t003:** Secondary metabolites with direct and indirect-antimalaria properties.

N	Compound	Biological Effect	Pharmacodynamics	Pharmacokinetics	References
**1**	α-Amaryn	- Immune-modulating properties;	N	N	[94]
		- Direct anti-malarial properties.			
**2**	β-Amyrin	- Immune-modulating properties;	N	N	[94]
		- Direct anti-malarial			
**3**	Apigenin	- Direct anti-malarial properties.	- Efflux of GS from mature erythrocyte.	N	[81]
**4**	Arteannoides	- Inhibition of TNF-α production - Anti-inflammatory.	- Inhibition of PGE2;	N	[26]
			- Inhibition of NO production.		
**5**	Artemisinic acid	- Weak anti-malarial properties.	N	N	[95]
**6**	Artemetin	- Weak anti-malarial properties.	N	N	[79]
**7**	Artemisinin (Qinghaosu)	- Direct anti-malarial properties;	- Inducer of CYP3A4;	- Metabolized by CYP3A4 and CYP2B6.	[73,96]
		- Anti-inflammatory;	- Inhibition of TNF-α gene expression.		
		- Inhibition of pro-inflammatory cytokines.			
**8**	Artemisitene	- Direct anti-malarial properties.	N	N	[95]
**9**	Artennuin B	- Weak anti-malarial properties;	- Inhibitor of CYP3A4.	-	[93,97]
		- Anti-inflammatory;			
		- Inhibition of IL-1β, IL-6 and TNF-α production.			
**10**	Artennuic acid	- Weak anti-malarial properties;	N	N	[80]
		- Anti-inflammatory.			
**11**	Artemisia ketone	- Direct anti-malarial properties;	- Inhibition of CYP2A6 CYP2B6	N	[88]
			- Inhibition of hemozoin crystallization.		
**12**	Casticin	- Indirect anti-malarial properties;	N	N	[79]
**13**	Chlorogenic acid	- Anti-inflammatory;	N	N	[34]
		- Inhibition of IL-8 and IL-6.			
**14**	Chrysosplenetin	- Weak anti-malarial properties.	N	N	[79]
**15**	Chrysosplenol-D	- Weak anti-malarial properties.	N	N	[79]
**16**	Cirsilineol	- Weak anti-malarial properties.	N	N	[79]
**17**	1α,4α-Dihydroxybishopsolicepolide	- Direct anti-malarial activity.	N	N	[14]
**18**	9-Epi-artemisinin	- Direct anti-malarial properties.	N	N	[95]
**19**	Eupalitin	- Anti-inflammatory.	N	N	[98]
**20**	Eupatorin	- Weak anti-malarial properties.	N	N	[79]
**21**	Friedelin	- Immune-modulating and anti-malarial activities.	N	N	[94]
**22**	Jasceolidin	- Anti-inflammatory.	N	N	[98]
**23**	Kaempferol	- Direct anti-malarial properties;	- Inhibition of GSK3β.	N	[83]
		- Anti-inflammatory.			
**24**	Luteolin	- Direct anti-malarial properties.	N	N	[80,99]
**25**	Mono caffeoylquinic acid	- Indirect anti-malarial properties.	N	N	[95]
**26**	Nerolidol	- Direct anti-malarial properties;	- Inhibition of CYP2A6 CYP2B6;	N	[88]
		- Anti-inflammatory.	- Inhibition of hemozoin crystallization.		
**27**	Quercetin	- Direct anti-malarial properties;	- Modulation of pro/Anti-inflammatory cytokines;	N	[76]
			- Inhibitor of thioredoxin reductase;		
		- Anti-inflammatory.			
			- Inhibition of GSK3β.		
**28**	Quercetin-3-galactoside	- Direct anti-malarial properties.	N	N	[34]
**29**	Quercetin-3-glucoside	- Direct anti-malarial properties.	N	N	[78]
**30**	Rhamnetin	- Direct anti-malarial properties.	N	N	[78]
**31**	Rosmarinic acid	- Indirect anti-malarial properties;	N	N	[34]
		- Anti-inflammatory;			
		- Inhibition of IL-8 and IL-6.			
**32**	Rutin	- Direct anti-malarial properties;	N	N	[78]
**33**	Scopoletin	- Anti-inflammatory;	- Modulation of Pro-/Anti-inflammatory cytokines	N	[80,100]
		- Indirect anti-malarial properties.			
**34**	Thujone	- Direct anti-malarial properties.	- Inhibition of CYP2A6 CYP2B6.	N	[88,89]
**35**	Tri-caffeoylquinic acids	- Indirect anti-malarial properties.	N	N	[95]
**36**	Yomogiartemin	- Direct anti-malarial activity.	N	N	[14]

N = No report. Compounds **1**, **3**, **4**, **5**, **6**, **7**, **8**, **25**, **32**, **33**, **34**, **35**, **30**, **31**, **27** and **28** isolated from *A. annua*. Compounds **2**, **9**, **10**, **11**, **12**, **13**, **14**, **15**, **16**, **17**, **18**, **19**, **20**, **21**, **22**, **23**, **24**, **26** and **29** isolated from both. Compounds **36** and **37** isolated from *A. afra*.

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
