# Peer review of "Secondary Metabolites Isolated from Artemisia afra and Artemisia annua and Their Anti-Malarial, Anti-Inflammatory and Immunomodulating Properties—Pharmacokinetics and Pharmacodynamics: A Review"

_metabolites, 2023, doi:10.3390/metabo13050613_

Round 1

Reviewer 1 Report (Previous Reviewer 3)

The paper by Shinyuy et al presents a comprehensive review of the chemical composition of two species belonging to the Artemisia plant family, namely A. annua and A. afra. The authors provide a detailed analysis of the therapeutic potential of several compounds present in these plants, including monoterpenes, sesquiterpenes, and polyphenols, that exhibit properties such as anti-malarial, anti-inflammatory, and immunomodulatory activities.

The authors report also an updated summary of the antimalarial properties of other Artemisia plant species.

The review expands the knowledge of Artemisia annua which is well-known as the plant producing artemisinin, a sesquiterpene lactone compound that has been used in traditional Chinese medicine for thousands of years and is nowadays the most effective drug for treating malaria in combination therapies (ACTs). As shown in this paper other plants of the same genus produce a large number of interesting compounds with varying pharmacological activities.  

The information summarized in this review is interesting, can be useful for many researchers. Therefore, I would recommend publication of the manuscript after revision

Points to improve before publication

The manuscript is a revised version of a previously submitted text. Although the present version has been improved some sections need further revision:

Abstract. Data in Table 2 describing the reported antimalarial properties of the different Artemisia spp are important and its presence in the manuscript should be mentioned in the abstract

Table 2. I would recommend reformatting thoroughly the information summarized in Table 2 to improve clarity

I suggest keeping an alphabetical order in each column:

·         Place Artemisia spp in alphabetical order

·         Next Plant part/extract also in alphabetical order and keep the same order throughout the table

This would greatly improve clarity

-Table 3. I suggest listing secondary metabolites in alphabetical order.  

-Figure 1. Needs improvement to look more ‘professional’ and keep up with publication standards

-Figures 2a-2d.

·         I would suggest placing structures in alphabetical order, for instance in Fig 2b Artemisia ketone, Nerolidol and Thujone

·         and then try to unify Figs 2a-2d into a single figure keeping the same size and format

-References. References also should be thoroughly reviewed. For instance

·         Ref 2-page numbers are repeated.

·         Ref 4 is incomplete.

·         Ref 117 is incomplete and probably many others

·         Capital letters are used in some references and should be avoided.

·         Some DOI are incorrect.

This is important because 202 references are included in the text and in their present form access to the information is difficult

In my opinion the manuscript is interesting and could be cited many times in the future, if carefully reviewed and modified.

Author Response

Thank you for your comments and suggestions to make the manuscript better. Please find in the attached file (pdf) the responses to the comments made.

Reviewer 2 Report (Previous Reviewer 2)

The resubmitted manuscript has been improved and the authors have adequately addressed my concerns with this manuscript. However, mistakes should be revised properly.

Typo

- Line 100: wed-data bases → web databases

- Line 101: Research Gate → ResearchGate, Science Direct → ScienceDirect

- Table 2: To my knowledge, the same number of significant figures should be applied in the text or table. Zero(0) at the hundredths needs to be removed or to be provided. Please make sure that the remaining numbers be correct.

- Lines 266, 421: "pro and anti-inflammatory" → "pro- and anti-inflammatory"

- All "Kg" should be corrected to "kg".

- Which one is correct? "mins" (line 442) or "min" (line 450)

- Which one is correct? "L/hr/kg" (line 449) or "l /h/kg" (line 450)

- Line 338: "CYP 450" → "CYP450"

- Line 410: "2.3h" → "2.3 h" or "2.3 hr"

Author Response

Thank you very much for theses observations especially in the typos. All comments have been effectively corrected as in the attached pdf file and in the manuscript.

Thanks. 

Reviewer 3 Report (New Reviewer)

The manuscript "Secondary Metabolites Isolated from Artemisia afra and Artemisia annua and their Antimalarial Properties. Pharmacokinetics and Pharmacodynamics. A Review.", shows relevant information on the two species. However, authors should consider the following points:

- The authors mention different activities (anti-inflammatory, antiplasmodic and immunomodulatory). However, the title of the manuscript does not reflect its content. I suggest the authors to modify the title.

- There is relevant information on A. afra, if the authors search for information with its other synonymous names (A. ponticum, tenuifolium, altaica, balsamita, grandiflora, pallida, pontica, pseudopontica and tenuifolia).

- There is relevant information on A. annua, if the authors search for information with its other synonymous name (A. chamomilla).

- Regarding the distribution of these two species, it would be helpful if the authors include a geographical map highlighting the places where they are found in greater proportion, are native or introduced.

- Put all the structures of the compounds of the species A. afra and A. annua. Likewise, all structures must have the same dimensions and the same format.

- When the antimalarial, anti-inflammatory and immunomodulatory activities of the compounds are reported, they must be separated as they were done with the structures with antiplasmodic activity. Since it causes confusion.

- If the structures of the compounds are going to be listed, they should be separated by activities and not only focus on the antiplasmodic.

- Although the authors describe a collection of data on the biological activities, they only emphasize the anti-inflammatory activity. The antiplasmodic and immunomodulatory activities are poorly detailed. The authors must indicate the mechanisms of action of the compounds on these activities.

-Authors must include the section on toxicity of extracts and compounds identified in these species

Author Response

Thank you very much for the comments and suggestions to make the manuscript better. In the attached file (pdf file) is the responses to the various points raised.

Thanks

Round 2

Reviewer 1 Report (Previous Reviewer 3)

The revised version of the paper by Shinyuy and colleagues has significantly improved with the incorporation of the comments and corrections suggested, and in my opinion, it is now almost ready for publication. This review discusses the antimalarial properties of secondary metabolites isolated from Artemisia species. It will make a valuable contribution to the field of malaria research, providing valuable information for researchers and clinicians alike.

 Minor points: I would still recommend that the authors make a minor modification to Table 1 and Table 2 for consistency. In Table 1, Artemisia species appear in abbreviated form (e.g., A. annua, A. afra), whereas in Table 2, they are written in full (e.g., Artemisia annua).

The quality of Figure 2 could still be improved. The authors should align the capital letters placed below each picture and ensure that all pictures have the same size and are aligned properly

Author Response

Dear Reviewer,

Thank you very much for these further comments. There have been effectively corrected as indicated.

Comments: I would still recommend the authors make minor modification to table 1 and table 2 for consistency. In table 1 artemisia spp appear in abbreviated form (e.g, A. annua, A. afra) whereas in table 2 there are written in full (e.g., Artemisia annua).

Response: This has been corrected. Table 1, Artemisia spp are now written in full (e.g., Artemisia annua) to match the form in table 2.

comments: The quality of figure 2 should still be improveD. The authors should align the capital letters placed between each figure and ensure that all pictures have the same size and are  aligned properly.

Response: The pictures have modified, aligned properly and the capital letter between aligned also.

Reviewer 3 Report (New Reviewer)

The authors have made the pertinent corrections made by the reviewers. The manuscript in its current state can be published on the journal's platform.

Author Response

Dear reviewer,

Thank you for your comments and the acceptance for publication.

Comments: The authors have made the pertinent corrections made by the reviewers. The manuscript in its current state can published in the journal's plat form.

Response: Thank you for your corrections and suggestions to make the work better.  

This manuscript is a resubmission of an earlier submission. The following is a list of the peer review reports and author responses from that submission.

Round 1

Reviewer 1 Report

Shinyuy et al. propose a review on Artemisia annua and A. afra, with a focus on the secondary metabolites and their antimalarial and anti-inflammatory properties. The subject is interesting and a bibliographic point is welcome. However, much of the information provided is approximate with inadequate or missing references. In particular, the authors state throughout the review that these two species are widely consumed around the world for their properties, especially in malaria endemic countries, but no works reporting the uses of these two plants outside their endemic areas are cited. Many reviews or point of views are cited, while research results are expected to support the claims made. They are also several misunderstanding of results (often taken from reviews) which are reflected into erroneous statements that are unacceptable. There are two many misinterpretation or missing information to trust the authors. Overall, the quality of this paper is poor, I do not recommend to accept it for publication.

My detailed comments are as follows:

-        References: a strong effort is needed on the form to homogenise and respect the instructions to the authors for the bibliographic listing and the citations in the text.

-        In the Abstract: sentence lines 36-38, you speak about “compounds involved in pharmacokinetics and pharmacodynamics properties”. I find this formulation clumsy because every compound has pharmacological properties that define how the drug is absorbed, distributed, metabolized, and excreted by the body. So, you’ve made the distinction between compounds involved in pharmacokinetics and pharmacodynamics properties and compounds not involved ? It makes no sense. Actually, you have made a focus on the main compounds that were detected in plasma and which have been the subject of pharmacokinetic studies.

-        Introduction lines 55-59, “the situation is … Africa”. When you speak about “antimalarial drug resistance”, you need to specify “parasite antimalarial drug resistance”. There is no link between parasite resistance to antimalarial drugs and HRP2 gene deletion. HRP2 gene deletion leads to a failure to detect parasite by rapid tests based on the HRP2 antigen, which is a very serious problem but independent of antimalarial resistance…

-        Introduction line 74 “more than 600 secondary metabolites” is not true, this “almost 600 secondary….”

-        Introduction lines 74-77: you claim that A. annua and afra are being consumed more as due to increased parasite resistance to antimalarial drugs in areas of high malaria endemicity. Do you have some reference please reporting a huge use of both plants in malaria endemic areas (others than those where the plants are endemic?)? If not, such claim can’t be written.

-        Introduction line 79: You need to replace “this resistance problem” which is not precise by “the emergence of artemisinin-resistant parasites.

-        Introduction lines 79-81: you speak about “emergence of the use of A. afra”, again no reference is cited to support this claim. Idem for “[A. afra] kills the malaria parasite”. The two references cited need to be completed with works demonstrating the antiplasmodial activity of A. afra on other stages than gametocytes, and with studies demonstrating the increasing uses of A. afra in malaria endemic areas.

-        Botanical paragraph lines 102 and 116-117 “treat a wide range of disease traditionally” and “due to its high medicinal potential”: add references.

-        Phytochemical paragraph: reference 20 is cited in the text but not in the Table 1 where the reference 30 is cited for the corresponding information, please homogenise.

-        Phytochemical paragraph line 160 “despite its multi-biological activity against infections”: add references.

-        Table 1: the last compound on page 5 has no reference.

-        Table 1: artemisin is called “Qinghaosu”, which is not a problem but the following Tables refer it as “artemisinin”, so please explain or homogenize.

-        Paragraph 4 on biological properties: Reference 41 is a review with no direct link with the sentence. This review presents the methods for studying compounds from plant extracts.

-        Paragraph 4 on biological properties lines 170-172: again, the authors claim that A. annua is used “across the globe” for the treatment of malaria, especially in high endemicity areas. No reference is cited.

-        Paragraph 4 on biological properties lines 172-173: “Tea infusions of A. annua is traditionally used to treat malaria”, please precise “in China” as the reference 42 cited refers to a paragraph where the authors report the traditional preparation use in China in the fourth century.

-        Lines 178-180: you begin here to speak about clinical trials, then you speak about in vitro data, and come again on clinical trials lines 187-198. Please group the clinical data together after in vitro data. Why citing here the reference 48 which reports preliminary clinical data ? Citing only the review 46 is more logical here, as an introduction to the paragraph on clinical data. The sentence should therefore be moved.

-        Lines 180-182: the work cited in ref 49 reports a very low IC50 value, but this paper does not explain the mode of calculation which probably reflects the IC50 of artemisinin contained in the in the extract studied. The tea was prepared with 9g of plant (instead of 5g more usually). The authors measured artemisinin content in two extracts and IC50s are presented in nM although in the method section they cite the WHO for classifying antiplasmodial activity according to ranges of IC50s values in µg/mL. This is really not enough precise to write this IC50 value without any explanation. De Donno et al published in 2012 a good paper presenting clearly antimalarial activity of A. annua tea with values for the tea in µg/mL and artemsinin IC50 values in nM based only on the artemisinin contained in the tea. There are many others papers studying IC50s of different type of extracts with varying results that should be reported here to have an idea of the variability of values. Table 2 could be cited here.

-        Lines 202-206: this point of view deserves argumentation and explanation.

-        Line 207: you speak about CYP2B6 and CYP3A4 cells which do not exist. The work was done to study the activity of CYP2B6 and CYP3A4 enzymes in HepaRG cells.

-        Line 223: The work cited in ref 26 demonstrated activity against early stage gametocytes only, not on late-stages as the results were not significant.

-        Line 250: “at the molecular” level?

-        Lines 255-258 needs references.

-        Line 273: again, “more than 600 metabolites” is not correct”, almost is more true.

-        Table 3: the information about cytokine inhibition should be written in the “biological effect” column. References 97-101 are not cited in the text, so that the information provided is not enough detailed.

-        Line 399: replace “dosage” by “formulations”.

-        Line 400: the ref 46 is a review although a research work is needed here. The work that demonstrated this should be cited here.

-        The content of paragraph 7 is light compared to what the title announces.  Half of it is devoted to a reminder of the parasitic mechanisms of artemisinin resistance.

-        Line 469: the sentence refers to clinical trials carried out with A. afra. Could you develop and give some references?

-        References are to fully revise, a few examples of : ref 117, no journal indicated; Ref 64 and 66 are the same; sometimes all the authors are listed, sometimes only one or a few; many are wtitten in capital letters, etc

-        Supplementary material: only Supplementary 4,5,6 are cited in the text, why add the other suppl materials?

-        In the middle of Suppl material, we find a Table 9…

Reviewer 2 Report

The purpose of this manuscript is to introduce the latest trends in secondary metabolites obtained from Artemisia sp. and their anti-malarial properties. This manuscript was described relatively well, and the contents would be informative to researchers that are trying to investigate anti-malarial compounds. However, few major points should be addressed and revised properly in the manuscript.

Major comments

  • The authors described the methods to collect the data in the abstract only, such as PubMed and Phytochemical and Ethnobotanical Databases. But there is no information in the main text of the manuscript. Please describe all of the databases and search keywords that they used.
  • I would like to recommend that the authors should focus on describing anti-malarial properties, not anti-inflammatory properties (lines 242-254, lines 331-343). In the current version, two properties are not related or complementary each other.
  • The authors use direct and indirect terms. But those terms are unclear. Please clarify the meaning.

Minor comments

  • line 58: Please provide the full name of the gene/protein in the first appearance, such as HRP2

Typos

  • Please choose one, anti-malarial or antimalarial.
  • line 49: one were -> one was
  • line 61: Please add "(WHO)"
  • line 95: China -> Chinese
  • line 138 and more: Arteanoides -> Arteannoides
  • Table 1: 1) Redundant "A. annua" at the column "Plant"; 2) If necessary, all of the first letters should be capitalized.
  • line 176-177 and more: and or -> and/or
  • line 179: has -> have
  • line 181 and more: antimalaria ->  anti-malarial
  • line 192 and more : Please add a space between number and unit. "5g" -> "5 g"
  • line 199: P450's -> P450
  • line 221: Tea -> tea
  • line 227 and more: Abbreviations should be spelled out in full at first; and after the first appearance, the abbreviation should be used. (Plasmodium -> P.) But please spell out in full at the beginning of the sentence.(lines 169: it's right)
  • line 229: Focused -> Focus
  • line 234: "A. indica A. maciverae" -> "A. indica and A. maciverae"
  • line 234: table -> Table
  • Table 2: Please check several typos, such as unnecessary space ("Leaves/ DCM") and necessary subscript ("H2O")
  • line 253, if necessary: NO -> nitric oxide (NO)
  • line 253: IL -> interleukin (IL)
  • line 257: herbs alba or herbs-alba?
  • line 258: "A. argyi, A. princeps" -> "A. argyi, and A. princeps"
  • line 264: spp -> spp.
  • line 264: ant -> anti
  • line 288: P. berghei infected -> P. berghei-infected
  • line 297: Plasmodium -> Plasmodium sp.. or Please clarify the specific species names.
  • line 302: 1) Please spell out the full name of ABCC1. 2) an erythrocytes ABCC1 mediated mechanism -> ABCC1-mediated mechanism
  • line 303: "erythrocyte-infected P. falciparum" -> "P. falciparum-infected erythrocyte" or  "erythrocyte infected with P. falciparum"
  • line 311: Why it the beta italicized?
  • line 317 and more: cytochrome P450 -> CYP
  • line 329: "Thujone, norilodol, artemisia ketone" -> "Thujone, norilodol, and artemisia ketone"
  • line 336, if necessary: nitric oxide (NO) -> NO
  • Table 3: Please 1) remove all periods; 2) capitalize the first letter of each column name; 3) explain the meaning of the hypen (no report or no metabolite); 4) remove either 27. Eupatorin or 31. Eupatorin.
  • line 347: "26. 29 " -> "26 and 29"
  • line 351: gamma -> ?
  • line 372: beta -> ?
  • line 405: Please provide specific time after oral administration.
  • line 410, if necessary: "pro and anti-" -> "pro- and anti-"
  • line 413: it slow -> its slow
  • line 439: l/h/kg -> L/hr/kg
  • line 460: revelled -> revealed
  • line 480: "B, (" -> "B ("
  • line 481: ") quercetin" -> "), quercetin"
  • line 488 and more: spp -> spp.
  • line 499: artemisia -> Artemisia
  • Supplementary 1: Please check several typos; unnecessary space ("108.  p-Menth-" and more), necessary italic type (Cis, Trans, Iso), capital letter (36. 4-terpineol, 51. azulene, and more). Please complete No. 150 name. Please remove redundant "[" at No. 39, [[126]

  • Supplementary 2: Same as commented above in Supplementary table 1.
  • Supplementary 3: Same as commented above in Supplementary table 1.
  • Supplementary 4: Same as commented above in Supplementary table 1. In No. 509, B or beta?
  • line 1034: Talbe 9 or Supplementary 6?
  • Supplementary 10: Please complete No. 599 name.

Reviewer 3 Report

The paper by Shinyuy et al is a review describing the phytochemical composition of two Artemisia species, A. annua and A. afra, and their potential pharmacological activities. It provides an extensive overview of compounds isolated from both species, including monoterpenes, sesquiterpenes, and polyphenols with anti-malarial properties, anti-inflammatory potentials and immunomodulating properties. The review expands the knowledge of Artemisia annua which is well-known as the plant producing artemisinin, a sesquiterpene lactone compound that has been used in traditional Chinese medicine for thousands of years and is nowadays the most effective drug for treating malaria in combination therapies (ACTs). As shown in this paper A. annua another plant of the same genus A. afra produce a large number of interesting compounds with varying pharmacological activities.  

The information summarized in this review is interesting, can be useful for many researchers and would become a reference in this field. Therefore, I recommend publication of the manuscript after revision

Points to improve before publication

The manuscript is well written and clear however some sections need revision:

-Table 3

Review carefully the information under the column Biological effect: some compounds are described as antimalaria others as antimalarial other antimalarial activity others as anti-plasmodial. Correct and unify

-References. References also should be thoroughly reviewed. For instance

·         Ref 2-page numbers are repeated.

·         Ref 4 is incomplete.

·         Ref 117 is incomplete and probably many others

·         Capital letters are used in some references and should be avoided.

·         Some DOI are incorrect.

This is important because 192 references are included in the text and in their present form access to the information is difficult

Figures 2a,2b,2c, 2d replace anti-plasmodial property and write anti-plasmodial activity